# A Biocontrol Strain of *Pseudomonas aeruginosa* CQ-40 Promote Growth and Control *Botrytis cinerea* in Tomato

**DOI:** 10.3390/pathogens10010022

**Published:** 2020-12-31

**Authors:** Xingyuan Wang, Xinan Zhou, Zhibo Cai, Lan Guo, Xiuling Chen, Xu Chen, Jiayin Liu, Mingfang Feng, Youwen Qiu, Yao Zhang, Aoxue Wang

**Affiliations:** 1College of Life Sciences, Northeast Agricultural University, Harbin 150030, China; wxy18246718896@163.com (X.W.); Xinan15245116526@163.com (X.Z.); Caizhibo2019@outlook.com (Z.C.); mingfangfeng@126.com (M.F.); yw12_630@126.com (Y.Q.); 2College of Horticulture and Landscape Architecture, Northeast Agricultural University, Harbin 150030, China; guolan1594637@163.com (L.G.); chenx@neau.edu.cn (X.C.); cxsteam@163.com (X.C.); 3College of Sciences, Northeast Agricultural University, Harbin 150030, China; 13040216@163.com

**Keywords:** tomato gray mold, *Pseudomonas aeruginosa*, *Botrytis cinerea*, biological control, growth promotion

## Abstract

*Botrytis cinerea* infection can be very devastating for tomato production, as it can result in a large-scale reduction in tomato fruit production and fruit quality after harvest. Thus, it negatively affects tomato yield and quality. In this study, a biocontrol bacteria CQ-4 was isolated and screened from the rhizosphere soil of tomato plants. Morphological, physiological, and biochemical characteristics and 16S rDNA sequence analysis revealed that it belongs to the species *Pseudomonas aeruginosa*, which has a strong antagonistic effect against *Botrytis cinerea*. In addition, the bacterium’s antibacterial spectrum is relatively extensive, and antagonistic tests have shown that it also has varying degrees of inhibition on other 12 plant diseases. The growth promotion test showed that the strain has a clear promotion effect on tomato seed germination and seedling growth. The growth-promoting effect on plant height, stem thickness, dry and fresh weight and main root length of tomato seedlings was significantly improved after the seeds were soaked in a bacterial solution of 2.5 × 10^8^ cfu mL^−1^ concentration. This did not only maintain the nutritional quality of tomato fruits, but also prevents them from rotting. In vitro and pot experiments showed that the strain CQ-4 can effectively control tomato gray mold, and the control effects on tomato leaves and fruits reached 74.4% and 66.0%, respectively. Strain CQ-4 induce plants to up-regulate the activities of four disease-resistant defense enzymes. The peak enzymatic activities of Phenylalanine Ammonia Lyase (PAL), polyphenol oxidase (PPO), peroxidase (POD), and Superoxide Dismutase (SOD) were increased by 35.6%, 37.6%, 46.1%, and 38.4%, respectively, as compared with the control group. This study found that the strain can solubilize phosphorus, fix nitrogen, and produce cellulase, protease, ferrophilin, and other antibacterial metabolites, but it does not produce chitinase, glucanase, and HCN (hydrocyanic acid). This research screened out an excellent *Pseudomonas aeruginosa* strain that can stably and effectively control tomato gray mold, and it provided theoretical basis for further development and the application of biological agents.

## 1. Introduction

Tomato (*Solanum lycopersicum*) is cultivated all over the world [1,2]. It is nutritious, aids digestion, can be used as an antioxidant, and improve immunity [3]. *Gray mold* is a worldwide disease that is caused by *Botrytis cinerea Pers.ex Fr*. of the genus Botrytis [4]. It has been reported that it infects more than 500 host plants, including tomatoes, strawberries, grapes, cucumbers, and many other important economic crops [5,6]. Leaves and fruits are the main parts of tomato that are affected by *gray mold* [7]. Infection usually starts from the tip of the leaf and advances inward along the veins until the leaves die, which leaves the surface with a grayish white mold layer [8]. In fruit, stigma and petals are infected first and then expanded to the stalk and surface of the fruit with more serious infection on the green part. Subsequently, a thick *gray mold* layer grows on the peel, which gradually causes the fruit to become soft and rotten [9]. *Botrytis cinerea* is a typical necrotrophic fungus that produces many pathogenic metabolites that kill the host cells and seize nutrients, which causes plant tissues to rot, wilt, and die [10]. The pathogenic mechanism of the disease is very complicated, mainly due to the production of cell wall degrading enzymes, plant endogenous hormones (such as ABA (abscisic acid)), small molecular acids, and mycotoxins [11].

The control methods for tomato *gray mold* include breeding resistant varieties and spraying chemical and biological control agents [12]. Tomato breeding for *gray mold* resistance faces many difficulties due to the lack of resistant materials. Chemical control is the most effective method for preventing and controlling tomato *gray mold* in agricultural production [13]. However, most of the chemical agents are artificially synthesized and they are difficult to degrade. Long-term and large-scale application has caused severe environmental pollution and disrupted ecological balance while residues on fruits and vegetables can lead to human and animal poisoning. Biological control on the other hand is the use of beneficial microorganisms for disease control. The beneficial microbe mounts pressure on the survival of plant pathogens, thereby reducing its pathogenic effect on plants. Biological control has little impact on the natural environment, it leaves no residues, and it is harmless to humans and animals [1]. Thus, it has become an ideal prevention method for disease control. Research and exploration have shown that dozens of antagonistic microorganisms, including fungi, bacteria, and actinomycetes, are useful in controlling tomato *gray mold* [14]. In recent years, a large amount of separation and screening work have been carried out in order to identify bacteria for the biocontrol of *Botrytis cinerea*. Bacteria in the genus Bacillus and Pseudomonas are widely developed and applied to control plant diseases [15]. Zhang et al. screened a strain of *Bacillus amyloliquefaciens* (fqhm-13), which has significant antagonism to *Botrytis cinerea* [16]. The results showed that the control effects of fqhm-13 and its fermentation supernatant on tomato fruit *gray mold* reached 69.0% and 78.1%, respectively. Qiao et al. found that, after irrigating tomato plants with *Bacillus subtilis* PTS-394, they can induce systemic resistance to *gray mold* and enhance plant immunity [17]. Gao et al. measured the control effect of *Pseudomonas chlorophylla* strain HL5-4 on tomato *gray mold* through in vitro and pot experiments [18]. The results showed that the control effect of HL5-4 strain against *gray mold* on isolated fruits and leaves reached 72.4% and 59.9%, and the control effect against tomato seedling *gray mold* was 62.9%.

Pseudomonas is a gram-negative bacterium that is widely distributed in rhizosphere soil and above ground of plants [19]. This kind of bacteria reproduces fast and is an ideal biocontrol agent for inhibiting plant diseases. The Palleroni method is widely used in the classification of Pseudomonas based on the group study of DNA-rRNA homology [20]. This research focused on the rRNAIhomologous group of *Pseudomonas fluorescens*, which mainly includes three species; *Pseudomonas aeruginosa*, *Pseudomonas fluorescens*, and *Pseudomonas putida* [21]. A large number of studies has shown that Pseudomonas has the ability to promote plant growth (such as dissolving phosphorus, potassium, and nitrogen), and produce antibacterial substances (such as antibiotics, extracellular hydrolase, and its secondary metabolites, etc.), thereby effectively inducing plant defense system against pathogenic bacteria [22]. The study found that *Pseudomonas aeruginosa* developed its biocontrol agent property through the production of antifungal derivatives, with the phenazine among them. The applications of crude phenazine that was synthesized by *Pseudomonas aeruginosa* UPMP3 and hexaconazole were comparatively evaluated for their effectiveness in suppressing basal stem rot infection in artificially G. boninense-challenged oil palm seedlings [23]. The purpose of this research is to screen and isolate biocontrol bacteria from the rhizosphere of plants that effectively inhibit *Botrytis cinerea* and identify the species of the isolated bacteria. The research also constructed the antibacterial spectrum and growth-promoting effect of the bacteria suspension on tomato seeds and seedlings. Its effect on the quality of post-harvest tomato fruits was also evaluated. In vitro and pot experiments were conducted in order to verify the control effect of this bacteria on tomato *Botrytis cinerea* and detect its effect on the activities of tomato leaf disease resistance-related defensive enzyme. This research provides potential strains for the development and application of biocontrol agents that are useful in preventing and controlling tomato *gray mold* and other diseases. 

## 2. Results

### 2.1. Isolation, Purification and Screening of Antagonistic Strains

Two hundred twenty-eight strains of bacteria were isolated and purified from the rhizosphere soil of tomato plants. After preliminary screening by plate confrontation, eight strains with varying degrees of inhibition against *Botrytis cinerea* were obtained. The strains after the initial screening were re-screened by the antibacterial circle method, and the plate confrontation test was carried out with the tomato gray mold, *tomato fusarium* wilt, and rice wilt pathogens. The results showed that strain 4 had the strongest inhibitory effect on tomato *gray mold* and it had a strong inhibitory effect on *tomato fusarium* wilt and rice wilt pathogen. Here, we named strain 4 as CQ-4 (Table 1, Figure 1).

The eight strains obtained in the preliminary screening were re-screened, and the plate confrontation test was carried out with the tomato gray mold, *tomato fusarium* wilt, and rice wilt pathogens. Subsequently, we use the width of the inhibitory zone to detect the inhibitory effect of each strain on the pathogen.

### 2.2. Identification of Strain CQ-4

#### 2.2.1. StrainCQ-4 Morphology Detection

The CQ-4 strain was streaked in LB medium and cultured at 28 °C for 24 h. The colony produced yellow-green pigment, spherical, moist surface, smooth, translucent, and neat edges (Figure 2a). When observed under a microscope, the bacteria are slightly curved rod-shaped, with different lengths. They were identified as gram negative and they have a single polar flagellum, without spores and capsules (Figure 2b).

#### 2.2.2. Molecular Identifification of Bacterial Strain CQ-4

The 16S rDNA sequence analysis results show that the 16S rDNA sequence is approximately 1441 bp in length, and CQ-4 is closely related to *Pseudomonas aeruginosa* JCM 5962^T^ (BAMA01000316) (Figure 3). Combining morphological observation, identifying physical and chemical characteristics and 16S rDNA gene sequence analysis, the strain CQ-4 was identified as *Pseudomonas aeruginosa*.

#### 2.2.3. Physicochemical Identifification of Bacterial Strain CQ-4

The physicochemical identification test results showed that the strain CQ-4 had fluorescent chlorophyll production, low salt tolerance, strictly aerobic bacteria, oxidized type, and Gram-negative. Methyl red reaction, V.P. test, H_2_S test, and indole test were all negative, while nitrate reduction reaction, starch hydrolysis, contact enzyme reaction, gelatin liquefaction, and citrate utilization tests were all positive (Appendix A). 

### 2.3. Determination of Antibacterial Spectrum of Biocontrol Bacteria CQ-4

The strain CQ-4 has varying degrees of antagonistic effects against 15 kinds of plant pathogens and a broad spectrum of inhibition. It had a better control effect than the control bacteria WZ-37. (Figure 4, Appendix A). In Appendix A, the biocontrol bacteria CQ-4 and the control strain WZ-37 were used in order to quantify the inhibition band width of various pathogens.

### 2.4. Determination of the Effect of Biocontrol Bacteria CQ-4 on Tomato Growth Promotion

The biocontrol bacteria CQ-4 has a growth-promoting effect on tomatoes. The experimental results show that, when CQ-4 suspension was diluted between10^3^ times (spore concentration 1.0 × 10^8^ cfu mL^−1^) and 10^4^ times (spore concentration 1.0 × 10^7^ cfu mL^−1^), it had a better growth-promoting effect on tomato seeds (Appendix A). Diluting the bacterial suspension of the biocontrol bacteria CQ-4 by 500 times (spore concentration 2.0 × 10^8^ cfu mL^−1^), 800 times (spore concentration 1.25 × 10^7^ cfu mL^−1^), and 1000 times (spore concentration 1 × 10^8^ cfu mL^−1^) promoted the growth of tomato seedlings. The highest radicle length of the CQ-4 treatment group was 1.52 times that of the CK (ddH_2_O treatment as mock) group, and the lowest was 1.30 times. The growth promotion effect was the highest when the spore concentration of strain CQ-4 was 1.0 × 10^8^ cfu mL^−1^ (Figure 5A).

After sowing the tomato seeds that were treated with the strain with the best dilution factor (spore concentration 1.0 × 10^8^ cfu mL^−1^), we measured the morphological indicators of tomato seedlings with a vernier caliper and a balance. The results showed that the biocontrol bacteria CQ-4 had apparent growth-promoting effects on the plant height, stem thickness, primary root length, fresh weight, and dry weight when compared with the control group (Figure 5B, Table 2).

We treated tomato fruits with the best dilution factor (spore concentration 1.0 × 10^8^ cfu mL^−1^) and observed bacteria appearance and physiological quality of tomato fruit. After 24 days of storage, the rot rate of tomato fruits that were treated with biocontrol bacteria was lower than that of the CK group with a decay rate of 32.4% and 53.2%, respectively, indicating that the strain CQ-4 has a good antiseptic effect (Figure 6A). With the extension of storage time, the weight loss rate of tomato fruits showed an upward trend. On the ninth day, the weight loss rate of each treatment group increased sharply. However, the rate of weight loss in the biocontrol bacteria CQ-4 treatment group was significantly lower than that of the CK group (Figure 6B).

Subsequently, we test the hardness, titratable acid, and soluble solid content of tomato fruits after storage. The results showed that, after 24 days of storage, the hardness of the biocontrol strains CQ-4, WZ-37 (A Biocontrol Bacterial Strain stored in the laboratory was used as the control) and CK group decreased by 38.3%, 25.4%, and 21.8%, respectively. Simultaneously, no significant differences were observed in the soluble solids and titratable acid content of each treatment group. The result indicates that the strain CQ-4 has no obvious adverse effect on the nutritional components of tomato fruits (Figure 6C).

### 2.5. The Biocontrol Effect of Strain CQ-4 on Tomato Gray Mold

In this experiment, the in vitro and greenhouse pot experiments were conducted in order to detect the biocontrol effect of strain CQ-4 on tomato gray mold. The results showed that this strain had a significant control effect of 74.4% on *Botrytis cinerea* infecting detached leaves, which was slightly higher than the control strain (WZ-37), but no significant difference when compared with that of 40% pyrimethanil (chemical) (Appendix A, Figure 7A). The strain CQ-4 had a 66.0% control effect on the fruit of *Botrytis cinerea*, which is higher than that of pyrimethanil (preventing gray mold) (Appendix A, Figure 7B).

The pot experiments showed that individual leaves began to develop disease on the fifth day after inoculation with *Botrytis cinerea*, and the disease index was 53.2% on the 20th day. The disease index of the tomato *gray mold* treatment group first inoculated with biocontrol bacteria and then inoculated was significantly lower than that of the CK group. The disease index of the first inoculated strain CQ-4 treatment group was 20.6%, and the disease prevention effect was close to that of the biological agent pyrimethanil. The disease index of the treatment group that was inoculated with tomato *gray mold* 1 day after the inoculation of the strain CQ-4 was 27.1%, which was higher than that of the treatment group that was inoculated with the strain CQ-4 first and then inoculated with tomato gray mold. The test results show that the control effect of the treatment group with pathogenic bacteria after inoculation with the strain CQ-4 is as high as 61.3%, which is better than the control bacteria WZ-37, and the control effect is close to the finished biological preparation (Table 3, Figure 8).

### 2.6. Detection of Defense Enzyme activities Induced by Biocontrol Bacteria CQ-4 in Tomato Leaves

#### 2.6.1. Effect of Phenylalanine Ammonia Lyase (PAL) Activity on Tomato Leaves after Treatment with Biocontrol Bacteria

The results of PAL enzyme activity showed that both CQ-4 treatment and under the stress of Botrytis cinerae could induce enhanced PAL activity in tomato leaves, and the inducing ability of CQ-4 treatment was stronger than Botrytis cinerea, reaching its peak on the fifth day, being 35.56% higher than CK group. The first inoculation of *gray mold* and the spraying of CQ-4 bacterial suspension treatment reached the peak on the third day, which was 34.73% higher than the CK group. It can be seen that the CQ-4 strain and *Botrytis cinerea* have a synergistic effect when induced together (Figure 9A).

#### 2.6.2. Effect of Peroxidase (POD) Activity on Tomato Leaves after Treatment with Biocontrol Bacteria

The POD activity of tomato leaves treated with CQ-4 was significantly increased in the absence of *Botrytis cinerae*, and it reached its peak on the fifth day. The POD activity of tomato leaves that were treated with CQ-4 was 46.05% higher than CK group. Subsequently, it decreased slowly, but it was still higher than the CK group (Figure 9B).

#### 2.6.3. Effect of Polyphenol Oxidase (PPO) Activity on Tomato Leaves after Treatment with Biocontrol Bacteria

The PPO activity of tomato leaves treated with CQ-4 could be improved in the absence of *Botrytis cinerae*, and it reached its peak on the fourth day, which was 37.63% higher than CK group, followed by a decrease. On the sixth day, it increased slowly, and then decreased again, but it was still higher than CK group. Under the stress of *Botrytis cinerae*, *Botrytis cinerae* inoculation reached a peak on the fifth day of CQ-4 suspension treatment, which was 30.55% higher than the CK group and then decreased, but still higher than the CK group (Figure 9C).

#### 2.6.4. Effect of Superoxide Dismutase (SOD) Activity on Tomato Leaves after Treatment with Biocontrol Bacteria

In the absence of *Botrytis cinerea*, CQ-4 could increase the SOD activity of tomato leaves, and it reached the peak on the fourth day, increased by 38.36% as compared with CK Under the stress of *Botrytis cinerea*, it reached the peak on the fifth day, increased by 27.63% when compared with CK, and then decreased, but was still higher than the control (Figure 9D).

### 2.7. Study on the Related Mechanism of Biocontrol Bacteria in Promoting Growth and Disease Resistance

We used plates to detect CQ-4 strains that are related to growth-promoting and disease-resistant activities. The results showed that the strain has the ability to decompose inorganic phosphorus, organic phosphorus, and nitrogen fixation, and it can secrete cellulase, protease, and the ability to bind Fe^3+^ with ferritin. However, it cannot produce chitinase and glucanase. For HCN (hydrocyanic acid) detection, the test paper did not change color, which indicated that HCN is not produced (Appendix A, Appendix A).

## 3. Discussion

Tomato *gray mold* is a ubiquitous and refractory disease that causes a decline in tomato yield and quality. It has a high incidence rate and can rapidly spread, therefore having a great impact on tomato yield. In severe cases, it can reduce the production by 40–50%, or even cease production [24]. The use of beneficial microorganisms and their metabolites in controlling tomato *gray mold* has become very important. In recent years, there have been reports on the application of *Pseudomonas aeruginosa* bacteria in the prevention and control of plant diseases. Viji et al. found that *Pseudomonas aeruginosa* has a strong resistance to stress through ultraviolet irradiation and it can significantly inhibit the gray spot disease of perennial ryegrass [25]. In this study, a strain of *Pseudomonas aeruginosa* (CQ-4) was isolated and screened from the rhizosphere soil of plants for a high antagonistic effect against *Botrytis cinerea*. We also found that this strain had varying degrees of antagonism against various plant disease pathogens.

In this study, the effect of strain CQ-4 on the growth of tomato seeds and seedlings was determined. The consistency in our results indicates that the strain CQ-4 has good biocontrol potential. The growth promotion and disease control effect of this strain will have a significant positive effect in tomato production. The biological control of devastating plant pathogens satisfies human demands for green and healthy agricultural products, because it leaves low residue on crops. Thus, we determined the effect of strain CQ-4 on the appearance and physiological quality of tomato fruits after harvest. We observed no significant difference between the indicators after the treatment of biocontrol bacteria and control treatment. It shows that it has no adverse reactions on the nutritional components of tomato fruits, and it meets the safety standards of production. Therefore, the strain CQ-4 could be used in tomato production to prevent and control tomato *gray mold* and for fruit preservation.

Much research has reported that biocontrol bacteria have a strong control effect on diseases in the laboratory conditions, but it is challenging to apply in actual crop production [26]. This is because the adaptability of biocontrol bacteria is low under field environment. In addition, it is easily affected by a variety of external factors, which makes the effect of biocontrol bacteria on plant diseases lower in the field than in the laboratory. In the greenhouse pot experiment, the biocontrol effect of CQ-4 on tomato *gray mold* was 61.3%. The result provides a theoretical basis for the biocontrol effect of the strain CQ-4 in the field test. It was observed that the control effect is not much different from that of the biological agent pyrimethanil. This result verifies that the control effect of the biocontrol bacteria CQ-4 on the disease is related to the artificial inoculation sequence. The result provides a theoretical basis for the biocontrol effect of the strain CQ-4 in the field test.

Existing studies have shown that growth promotion and resistance to disease induced by biocontrol bacteria through the induction of defense enzymes is similar to the mechanism of innate plant resistance. The increase of PAL activity contributes to the accumulation of lignin, the synthesis of phenolics and phytoalexins, and the activities of PPO, POD, and SOD also increase to varying degrees after inoculation with biocontrol bacteria. Therefore, in this experiment, the four enzyme activities were used in order to identify the effect of strain CQ-4 on the growth and disease resistance of tomato leaves. The results showed that the CQ-4 strain alone and under *gray mold* stress could increase the PPO, POD, and PAL activities in tomato leaves. Therefore, we hypothesized that the CQ-4 strain may induce disease resistance and promote growth by interacting with tomato *gray mold* through a systemic mechanism. However, further research is needed on the specific correlation between the control effect of the CQ-4 strain and the defense-related enzyme activity in tomato leaves.

Siderophore is the most common bacteriostatic metabolite that is produced by Pseudomonas biocontrol strains, which inhibit the reproduction of pathogenic microorganisms and development of plant diseases [27]. The surfactant rhamnolipid that is produced by *Pseudomonas aeruginosa* has good fungus-inhibiting activity and it could be used in the petroleum and pharmaceutical industries. In this study, the anti-bacterial substances related to disease resistance and the growth promotion of CQ-4 strain were measured. The results showed that strain CQ-4 could not produce chitinase, glucanase, and HCN, but it could decompose inorganic phosphorus, organic phosphorus, and nitrogen fixation, and secrete cellulase, protease, and ferritin.

The application prospect of *Pseudomonas aeruginosa* is extensive, and it is valuable in the agriculture, medicine, and cosmetics industries. Meanwhile, it is also a pathogenic bacterium with virulence to humans and animals. but not plants. Therefore, it is ideal to use *Pseudomonas aeruginosa* metabolites as microbial pesticides in order to prevent and control diseases [28]. *Pseudomonas aeruginosa* K2187 can inhibit the growth of 36 fungal diseases. Analysis of its components revealed that it can produce chitinase and lysozyme [29]. Studies have shown that *Pseudomonas aeruginosa* can produce many types of enzymes, such as proteases, dehydrogenases, and lipases, which can catalyze some esters or water-insoluble substances, and they are often used in industrial enzyme preparations [30,31]. In general, a considerable amount of research is still needed on the potentials of *Pseudomonas aeruginosa*, as there exist a number of unsolved problems. For example, there is a shortage of biocontrol agents that can be developed and utilized in production; live biocontrol strains have unstable effects in the field, and they are easily degraded when applied. A high fermentation cost also negatively affects the development and promotion of bio-based agents [32]. Therefore, further research on the antibacterial metabolites of *Pseudomonas aeruginosa* biocontrol strains is a top priority. At the same time, exploring the biocontrol mechanism of *Pseudomonas aeruginosa* in plants can also provide a theoretical basis for the mechanism of human and animal susceptibility.

## 4. Materials and Methods 

### 4.1. Materials 

Forty-five soil samples were collected from the rhizosphere parts of plants, such as tomatoes, cucumbers, kidney beans, bananas, and pine trees. The collected soil samples were separated and purified for biocontrol bacteria, streaked inoculated into test tube slope, and stored at 4 °C. The other 13 pathogenic bacteria, including *Botrytis cinerea*, are preserved in the Tomato Laboratory of Northeast Agricultural University. The biocontrol bacteria Bacillus velezebsis WZ-37 is deposited in our laboratory. The tomato variety “Dongnong 713” was provided by Jingbin Jiang from the Tomato Research Group of Northeast Agricultural University. The test medium was LB medium (5 g yeast extract, 10 g peptone, 10 g NaCl, 15–20 g agar, 1000 mL distilled water, pH 7) and PDA medium (20 g glucose, 200 g potato, agar 15–20 g, 1000 mL distilled water, pH 7).

### 4.2. Separation, Purification and Screening of Biocontrol Bacteria

Antagonistic strains were repeatedly screened using the inhibition zone method. A 5 mm diameter tomato *Botrytis cinerea* bacteria cake was placed in the center of the potato dextrose agar (PDA) culture medium, it and inoculated three different isolated and purified bacteria equidistantly at a distance of 3 cm from the center. The plates were then cultured at 28 °C for seven days, and whether it produces a transparent antibacterial band was observed. 

The strains that are obtained by the preliminary screening are made into a bacterial suspension, and then a plate confrontation test is carried out with round cakes of *Botrytis cinerea*, Fusarium wilt, and Rhizoctonia solani in order to measure the width of the transparent antibacterial zone. Each group was repeated three times.

### 4.3. Identification of Biocontrol Strain CQ-4

Morphological identification: cultured CQ-4 bacterial suspension was streaked on PDA medium plate, placed at 28 °C for 24 h, and colony characteristics, such as morphology, size, color, and transparency, were directly observed.

Physiological and biochemical identification: according to [33], the Mérieux VITEK 2 Compact automatic bacterial identification system was used in order to analyze the physiological and biochemical characteristics of strain CQ-4.

16S r DNA sequence analysis: the genomic DNA was extracted by the CTAB method [34], and PCR amplification was carried out with the bacterial universal primer 27F (5′-AGAGTTTGATCMTGGCTCAG-3′) 1492R: (5′-TACGGYTACCTTGTTACGACTT-3′) [35]. The PCR reaction system (20 μL) is template DNA 3.0 μL, dNTP Mixture (2.5 mM) 2 μL, 10 × LA Taq Buffer II 2 μL, upstream primer 1.0 μL, downstream primer 1.0 μL, and ddH_2_O in order to make up the remaining volume. The amplification program is 94 °C pre-denaturation 3 min., 30 cycles (denaturation 94 °C for 30 s, annealing 55 °C for 30 s, extension 72 °C for 1 min.), and then 72 °C extension for 10 min. The PCR products were recovered by gel, and the recovered products were handed over to Beijing Huada Gene Company for sequencing analysis. The sequence methodology is Illumina. Sequence analyses of 16S rDNA were performed with the program EZBioCloud (https://www.ezbiocloud.net/) [36]. The nucleotide sequences of the partial 16S rDNA gene have been deposited in the GenBank database under the accession numbers. The accession number in NCBI is JCM 5962 (BAMA01000316) of the 16s sequencing of the novel *Pseudomonas* isolate. The phylogenetic tree was constructed with MEGA 6.06 software [37]. 

### 4.4. Determination of Antibacterial Spectrum of Biocontrol Bacteria CQ-4

Cakes of 15 kinds of pathogenic bacteria with a diameter of 5 mm were placed in the center of PDA culture medium, and the biocontrol strain CQ-4 was placed at the upper and lower ends at a distance of 3 cm from the center. Finally, 10 μL of bacterial suspension was added for the flat-plate confrontation test. We use WZ-37 as a control bacterium, sterile water as a negative control, and culture at 28 °C for seven days. The width of the transparent antibacterial zone was measured with a vernier caliper and then repeated three times for each group.

### 4.5. In Vitro Control Effect of Biocontrol Bacteria CQ-4 on Tomato Gray Mold

The control effect test of detached leaves: sterilized tomato leaves were placed in a soaked two-layer sterile filter paper petri dish. Cotton wool was dipped in the same concentration (1.0 × 10^8^ cfu mL^−1^) of CQ-4, WZ-37 bacterial suspension and then smeared evenly on the leaves, 40% pyrimethanil was used as a positive control, sterile water as a blank control, and then dried. *Botrytis cinerea* was inoculated, cultured at a constant temperature of 25 °C, and observed daily in order to calculate the incidence of detached leaves, disease index, and control effect. Fifteen leaves were used for each treatment and then repeated three times.

In vitro fruit control effect test: skin of disinfected tomatoes fruit was peeled off and then sprayed with the same concentration (1.0 × 10^8^ cfu mL^−1^) of CQ-4, WZ-37 bacterial suspension, and 40% pyrimethanil as positive control, while sterile water was used as a blank control. After drying, *Botrytis cinerea* hyphae was inoculated, cultured at 25 °C, and observed the disease on a daily basis for seven days. There were five fruits for each treatment, two points for each fruit, and three repetitions. Appendix A classifies tomato *gray mold* disease and it calculates the incidence, disease index, and control effect according to the following methods. Incident rate (%) = Number of diseased leaves and fruits/Investigate the total number of leaves and fruits ×100. Disease severity was evaluated seven days after inoculation on a scale of 0 to 4. Level 0 stands for no disease, level 1 represents morbidity ≤25%, level 2 represents morbidity 25.1–50%, level 3 represents morbidity 50.1–75%, and level 4 represents morbidity ≥75.1%. There were three replications for each treatment. All 30 plants per treatment were used for disease symptom investigation. The plant disease index (DI), which would represent both disease incidence and symptom severity, can be calculated as: DI = (ΣDi × Dd)/(Mi × Md) × 100, where i means a 0–4 disease level and Mi means plant number of reaction i, and Dd means total number of leaves investigated [38]. 

### 4.6. The Effect of the Biocontrol Bacteria CQ-4 on the Greenhouse Potted Control of Tomato Gray Mold

Sterilized tomato seeds were sowed in a 12 × 6 hole plug tray, with water being cultivated for 15 days. The seedlings were then transplanted into nutrient bowls. The spore concentration of *Botrytis cinerea* on tomato leaves was 1.0 × 10^6^ cfu mL^−1^, and the spore concentration of biocontrol bacterial suspension was 1.0 × 10^8^ cfu mL^−1^. After various treatments were applied, they were cultured at 25 °C, 75% relative humidity, and disease was evaluated on the tomato plant on the 20th day. Thirty days later, the greenhouse control effect test was conducted. Damage level on leaves was evaluated and the disease index was calculated, according to the tomato *gray mold* grading standard.

### 4.7. Growth-Promoting Effect of Biocontrol Bacteria CQ-4 on Tomato Seeds and Seedlings

Same-size grain tomato seeds were selected and added to 2% sodium hypochlorite (v/v) for 15 min., 75% alcohol, 30 s treatment, and then washing three times in sterile water [39]. The experiment was divided into two times and ten treatments were designed: first, fresh CQ-4 strain and control strain WZ-37 were transferred to LB medium for 24 h at 28 °C and 180 r/ min. The suspension of the biocontrol bacteria CQ-4 was diluted in the order of 0 times (spore concentration 1.0 × 10^11^ cfu mL^−1^), 10 times (spore concentration 1.0 × 10^10^ cfu mL^−1^), 10^2^ times (spore concentration 1.0 × 10^9^ cfu mL^−1^), 10^3^ times (spore concentration 1.0 × 10^8^ cfu mL^−1^), and 10^4^ times (spore concentration 1.0 × 107 cfu mL^−1^). Sixty seeds were soaked in each dilution gradient for 4 h and the biocontrol bacteria WZ-37 was used as a positive control, while clean water was used as a negative control. Bacteria were cultured at 28°C under constant temperature and humidity, and the average radicle length was recorded six days later. The tomato seeds were soaked with spore concentration (1.0 × 10^8^ cfu mL^−1^) and WZ-37 (spore concentration 1.0 × 10^6^ cfu mL^−1^) for 4 h. The treated seeds were sown in pots and plant height, stem thickness, main root length, and dry weight of tomato seedlings were measured for 30 days. The average value of these parameters was computed and replicated three times.

### 4.8. Effect of Biocontrol Bacteria CQ-4 on the Quality of Postharvest Tomato Fruits

Tomato fruits of the same size and maturity without damage and disease were selected, washed with sterile water, and then weighed. The suspension of the biocontrol bacteria CQ-4 was diluted 500 times (spore concentration 6.5 × 10^9^ cfu mL^−1^). Tomato fruit was soaked in the dilution for 5 min., dried, and stored at room temperature (20 ± 2 °C) for 24 days. Biocontrol bacteria WZ-37 was used as a positive control, and sterile water as a negative control. The decay and physiological quality of the fruit was evaluated at three days interval, and weight loss was recorded. Decay rate (%) = number of rotted fruits/number of investigated fruits × 100%, weight loss rate (%) = (initial fruit weight–investigation fruit weight)/initial fruit weight × 100%. Fruit hardness was measured with GY-4 fruit hardness meter; soluble solids were measured with a handheld refractometer; and, titratable acid content is measured with acid-base titration [39].

### 4.9. Effect of Biocontrol Bacteria on Inducing Defensive Enzymes Related to Tomato Leaf Resistance

Well-growing tomato seedlings were selected as experimental material and divided into five treatment groups. The first group was sprayed with sterile water, the second group was sprayed with CQ-4 bacterial suspension (spore concentration 1.0 × 10^8^ cfu mL^−1^), and the third group was sprayed with tomato *Botrytis cinerea* suspension (spore concentration 1.0 × 10^6^ cfu mL^−1^). The fourth group was first sprayed with the tomato *Botrytis cinerea* suspension and then sprayed with the CQ-4 suspension, while the fifth group sprayed the CQ-4 bacterial suspension first and then sprayed the tomato *Botrytis cinerea* suspension. We sampled after seven days in order to determine the content of defense enzymes.

For enzyme extracts and assays, fresh tomato leaves (0.1 g) were ground in liquid nitrogen, and then suspended in 0.9 mL solution containing 10 mM phosphate buffer (pH 7.4). The homogenate was centrifuged at 4 °C, 10,000 rpm for 10 min. and the resulting supernatant was collected for the determination of the activities of Phenylalanine ammonialyase (PAL), peroxidase (POD), polyphenol oxidase (PPO), and superoxide dismutase (SOD) while using commercial assay kits purchased from Nanjing Jiancheng Bioengineering Institute (Nanjing, China). All of the enzymes above were detected using a microplate reader (SpectraMax M5, Shanghai, China), and five to 10 seedlings were used in order to provide enough amounts of leaf tissues in each experimental replicate (*n* = 3).

### 4.10. Determination of the Effects of Biocontrol Bacteria on Plant Growth and Disease Resistance

The determination of the ability of biocontrol strains to secrete extracellular enzymes: Strain CQ-4 was spotted on cellulase, β-1,3-glucanase, protease, and chitinase culture media, and then cultured at 30 °C for 48 h, media strained, and observed for a transparent circle. The presence of cellulase, β-1,3-glucanase, protease, and microbiological enzymes was then evaluated and measured by the diameter of the transparent circle.

The determination of the ability to dissolve phosphorous and phosphate: the phosphate-dissolving circle method was employed. The CQ-4 bacterial suspension was spotted on the Monkina organophosphate and PKO inorganic phosphorus medium. After incubating at 28 °C for 2–3 days, the presence of transparent circles was observed, and the experiment repeated three times.

The determination of nitrogen fixation ability: CQ-4 bacteria suspension was spotted on the nitrogen fixation test medium, cultured at 28 °C for two to three days, and observed whether there is a transparent circle and the experiment was repeated three times.

Test for the ability to secrete iron: Chromazurine CAS test medium was used, CQ-4 bacteria suspension was inoculated on the medium, and then placed at 28 °C for three days, and presence of an orange halo around the colony was observed. The experiment was repeated three times. The method described in Castric KF et al. was used in order to evaluate color development on test paper, and experiment was repeated three times [40].

## Figures and Tables

**Figure 1 pathogens-10-00022-f001:**
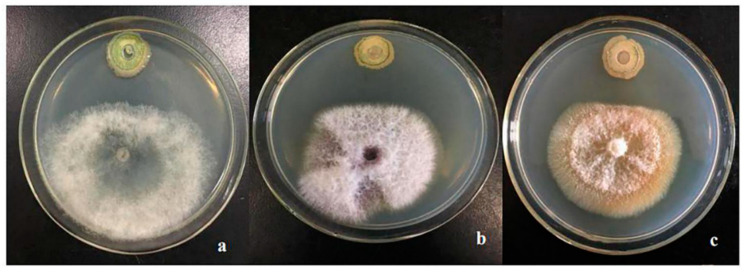
Antibacterial effect of CQ-4 on plant diseases (**a**). *Botrytis cinerea* (**b**). Tomato *Fusarium* Wilt (**c**). Rice Blight. Filter paper ring method was used to screen the preliminarily screened strains, and the plate confrontation test was conducted with the pathogens of *Botrytis cinerea*, Tomato *Fusarium* Wilt, and Rice Blight Damping off.

**Figure 2 pathogens-10-00022-f002:**
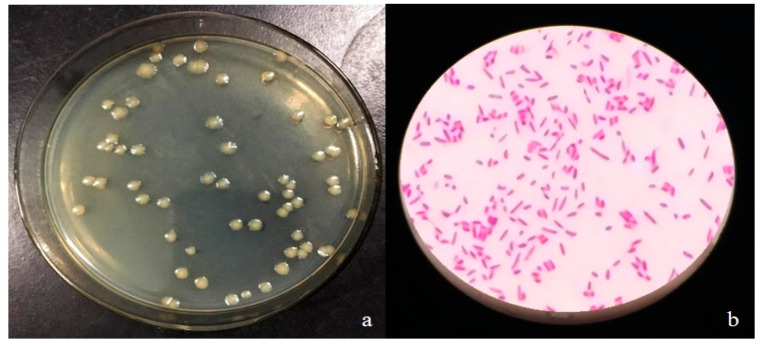
Morphological characteristics of strain CQ-4 on LB medium plate culture (**a**) and observed under an optical microscope (**b**).

**Figure 3 pathogens-10-00022-f003:**
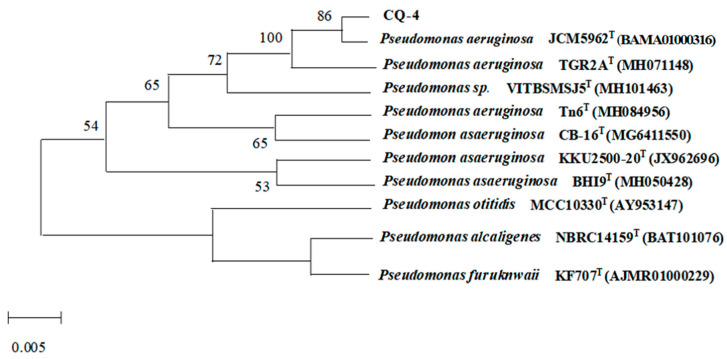
16SrDNA sequence phylogenetic tree of CQ-4. The nodes of the phylogenetic tree where the value of bootstrap is greater than 50%, which will be noted in the graph, and the superscript “T” indicates the model strain.

**Figure 4 pathogens-10-00022-f004:**
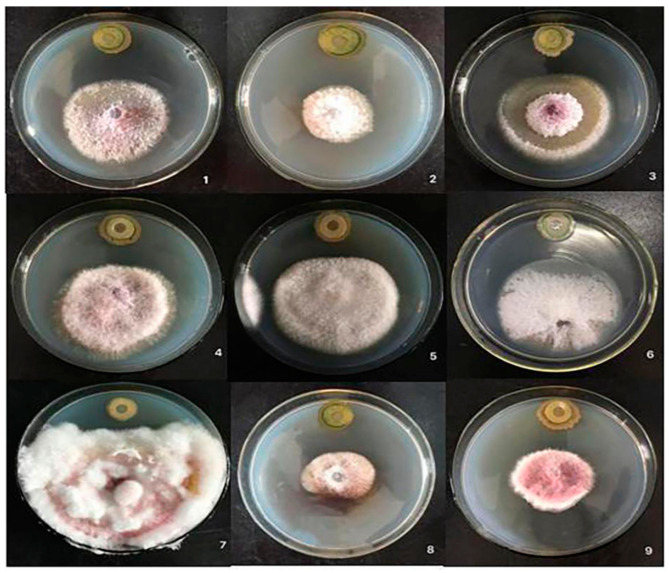
Determination of antibacterial spectrum of the biocontrol strain CQ-4. Note: 1. *Melon Fusarium* wilt. 2. *Sclerotinia sclerotiorum*. 3. *Watermelon Fusarium* wilt. 4. *Verticillium* wilt. 5. *Cucumber Fusarium* wilt. 6. *Cucumber anthracnose* pathogen. 7. *Fusarium graminearum* F1403. 8. *Bean anthracnose*. 9. *Acanthopanax* root rot.

**Figure 5 pathogens-10-00022-f005:**
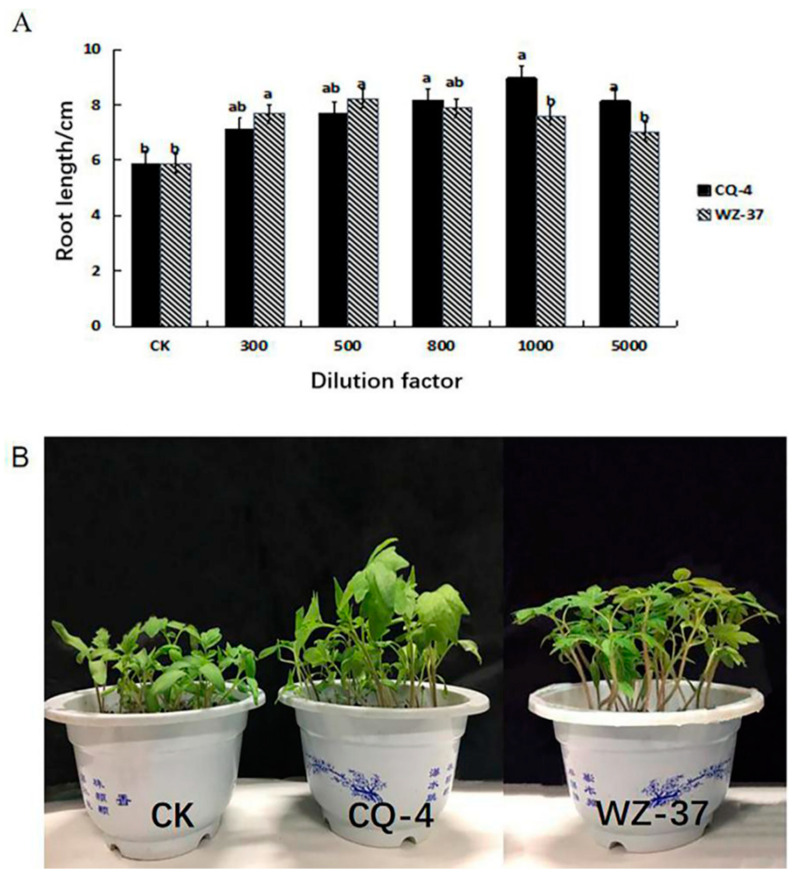
Determination of the Effect of Biocontrol Bacteria CQ-4 on the growth of tomato seed radicle and seedlings. (**A**) Effect of different dilutions of bacterial suspension on the growth of tomato seed radicle. (**B**) Effect of biocontrol bacteria on the morphological indexes of tomato seedlings. The different normal letters in the same point indicate significant difference among treatments at 0.05 level (*n* = 3).

**Figure 6 pathogens-10-00022-f006:**
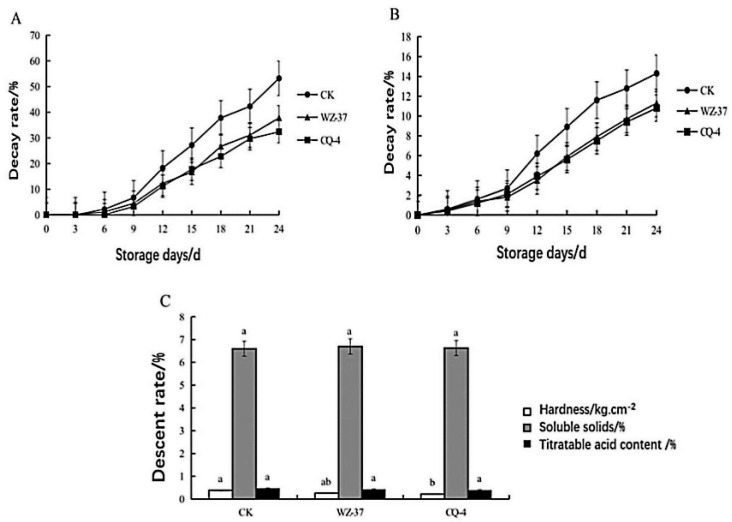
Determination of the Effect of Biocontrol Bacteria CQ-4 on decay rate, weight loss, physiological quality of tomato fruits. (**A**) Effects of biocontrol bacteria on decay rate of tomato fruits. (**B**) Effects of biocontrol bacteria on weight loss of tomato fruits. (**C**) Effect of biocontrol bacteria on physiological quality of tomato fruits. Decay rate (%) = number of rotted fruits/number of investigated fruits × 100%, weight loss rate (%) = (initial fruit weight–investigation fruit weight)/initial fruit weight × 100%. Fruit hardness was measured with GY-4 fruit hardness meter; soluble solids were measured with a handheld refractometer; titratable acid content is measured with acid-base titration. The different normal letters in the same point indicate significant difference among treatments at 0.05 level (*n* = 3).

**Figure 7 pathogens-10-00022-f007:**
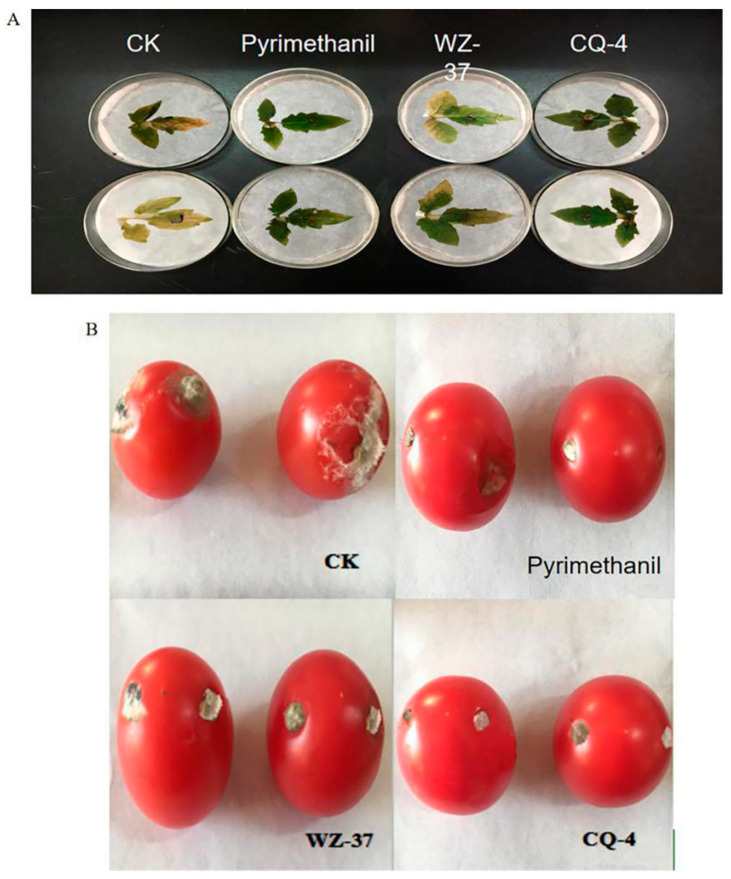
The Biocontrol Effect of Strain CQ-4 on Tomato Gray Mold in vitro experiments. (**A**) Effects of biocontrol bacteria on the in vitro leaves of tomato gray mold. (**B**) The effects of biocontrol bacteria on the in vitro fruits of tomato gray mold.

**Figure 8 pathogens-10-00022-f008:**
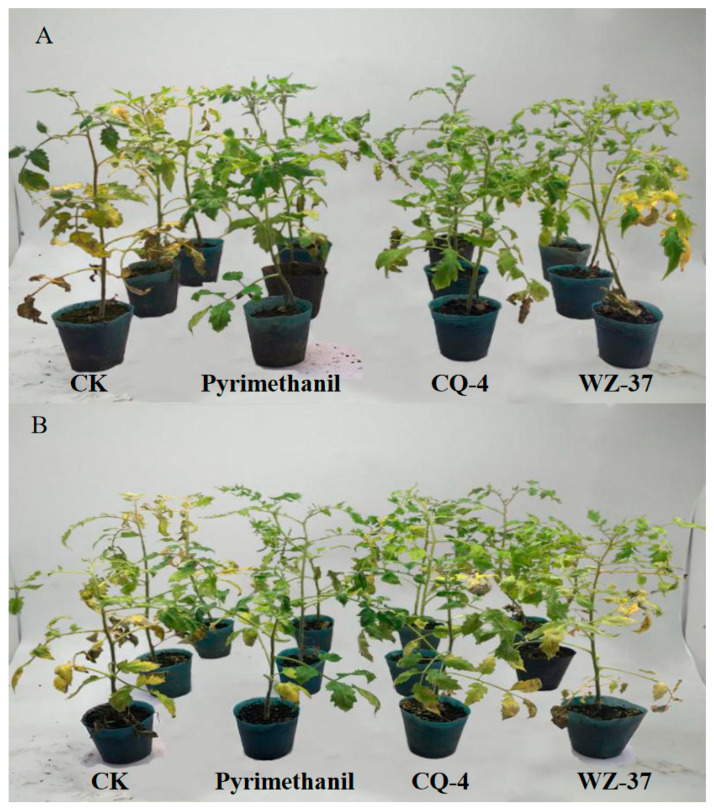
The Biocontrol Effect of Strain CQ-4 on Tomato Gray Mold in the greenhouse pot experiments. (**A**) Biocontrol bacteria preventive effect test. (**B**) Biocontrol bacteria treatment effect test.

**Figure 9 pathogens-10-00022-f009:**
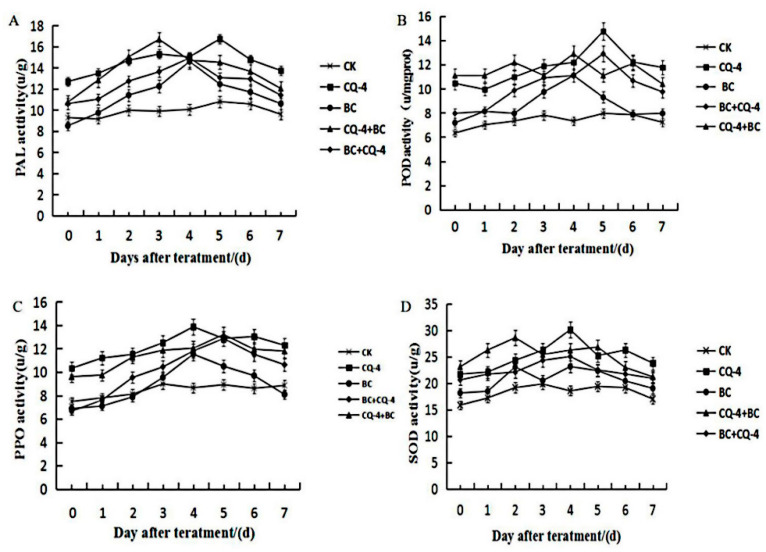
Detection of Defense Enzyme activities. (**A**) Effect of Phenylalanine ammonialyase (PAL) activity on tomato leaves after treatment with biocontrol bacteria. (**B**) Effect of Peroxidase (POD) activity on tomato leaves after treatment with biocontrol bacteria. (**C**) Effect of Polyphenol oxidase (PPO) activity on tomato leaves after treatment with biocontrol bacteria. (**D**) Effect of Superoxide dismutase (SOD) activity on tomato leaves after treatment with biocontrol bacteria.

**Table 1 pathogens-10-00022-t001:** Screening results of biocontrol bacteria.

Strain	Inhibition Zone Width/mm		
	*Botrytis cinerea*	*Fusarium oxysporum*	*Rhizoctonia solani*
2	5.6 ± 0.5 bc	1.8 ± 0.6 e	4.8 ± 0.7 b
4	9.6 ± 0.3 a	10.8 ± 0.2 a	9.3 ± 0.4 a
7	6 ± 0.4 b	2.1 ± 0.4 de	3.8 ± 0.3 bc
12	5.1 ± 0.3 c	2.3 ± 0.3 d	3.0 ± 0.3 c
19	4.1 ± 0.4 cd	3.8 ± 0.4 c	1.8 ± 0.6 cd
71	4.6 ± 0.1 c	3.0 ± 0.5 cd	4.1 ± 0.1 bc
133	3.8 ± 0.4 d	4.5 ± 0.2 bc	2.5 ± 0.2 cd
179	7.8 ± 0.2 ab	6.2 ± 0.5 b	1.3 ± 0.3 d

Notes: Different letters indicate significant differences within each category according to one-way analysis of variance (ANOVA) followed by Duncan’s test at the 0.05 alpha-level of confidence.

**Table 2 pathogens-10-00022-t002:** Effect of biocontrol bacteria on morphological indexes of tomato seedlings.

Treatment	Plant Height/mm	Thick Stem/mm	Taproot Length/mm	Fresh Weight/g	Dry Weight/g
CQ-4	80.04 ± 0.3 a	2.69 ± 0.1 a	88.59 ± 0.1 a	1.66 ± 0.1 a	0.17 ± 0.3 a
WZ-37	78.56 ± 0.7 ab	2.07 ± 0.1 b	71.82 ± 0.7 b	1.34 ± 0.4 ab	0.14 ± 0.2 b
CK	69.39 ± 0.8 b	1.68 ± 0.06 c	59.32 ± 0.6 c	0.72 ± 0.3 c	0.07 ± 0.1 c

Notes: Different letters indicate significant differences within each category according to one-way analysis of variance (ANOVA) followed by Duncan’s test at the 0.05 alpha-level of confidence.

**Table 3 pathogens-10-00022-t003:** Biocontrol bacteria control effect against *Botrytis cinerea*.

Treatment	Disease Index (%)		Control Index (%)	
	Biocontrol Bacteria + Pathogens	Pathogenic Bacteria + Biocontrol Bacteria	Biocontrol Bacteria + Pathogens	Pathogenic Bacteria + Biocontrol Bacteria
CQ-4	20.63 c	27.12 c	61.25 ab	49.06 b
WZ-37	24.71 b	28.06 b	53.58 c	47.29 c
40% pyrimethanil	18.97 d	24.88 d	64.36 a	53.26 a
CK	53.24 a	53.24 a	-	-

Notes: Different letters indicate significant differences within each category according to one-way analysis of variance (ANOVA) followed by Duncan’s test at the 0.05 alpha-level of confidence.

## Data Availability

Data available in a publicly accessible repository.

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
