# Peer review of "A Biocontrol Strain of Pseudomonas aeruginosa CQ-40 Promote Growth and Control Botrytis cinerea in Tomato"

_pathogens, 2020, doi:10.3390/pathogens10010022_

Round 1
Reviewer 1 Report
Your research found the novel strain of Pseudomonas aeruginosa which has a strong antagonistic effect against Botrytis cinerea. You compared the biocontrol activity with some strains and evaluated CQ-4.
Surprisingly, CQ-4 has as high effect as fungicide pyrimethanil. It interests me.
However, some information is lacked and I cannot evaluate your research completely.
You should read carefully and modify.
The presence of significant differences is described in the text. However, there is no explanation of the annotations in the figures and tables. Statistical methods are not also described. Please clarify.
In 2.4, 4.7 and 4.8, there is no standardized way of writing, such as by spore concentration or by number of dilutions. It confuse me to understand how bacterial concentration were effective.
Please standardize your writing style.
In 2.6 and 4.9, you measured the enzyme activities. The explanation is too complicated and difficult to understand.
The sampling or measurement condition is not described for detail, so I cannot judge whether the experiment is done correctly.
Are the samples contaminated with the treated microorganisms and their enzymes? Does contamination also affect the measured values?
If you cannot explain this section for detail, I suggest to delete this experiment.
minor points;
Overall. The name of species should be italicized.
L77. "a gram-negative"
L81. "3 species"
L141. what is "bacon"?
L141. what is "CK"? (maybe mock / negative control.)
Please describe the condition and the abbreviation.
L197 You should explain the abbreviation here, not later (L371).
L201 Is this sentence incorrectly inserted?
Reviewer 2 Report
The manuscript sent for review concerns the use of bacteria for the biocontrol of tomatoes against the development of gray mold. It is a pathogen that causes significant losses in the storage of tomatoes. The manuscript is interestingly written and the research is innovative.
Please clarify / improve certain issues:
line 53-54 can be expanded with specific examples of control methods, e.g. from doi.org/10.1007/s11947-020-02487-w
table 1 - names should be in italics
line 123 - Pseudomonas in capital letters
figure 3 - names in italics
figure 4A, 5C - no statistical description
line 363-372 - how were enzymes tested, was ready kit used?
References must be adapted to the journal's requirements
Reviewer 3 Report
The authors claim to have isolated a novel Pseudomonas aeruginosa strain, able to provide a prophylactic biocontrol for the fungus Botrytis cinerea in tomato plants. The authors provide evidence of agricultural characteristic advantages, in accordance with international literature. Isolation and characterization of endophytic or soil Pseudomonas species as biocontrol agents is a trending topic in phytopathology, mainly due to the ease and cheap use for manipulating these bacteria species and their significant relation against phytopathogenic fungus.
Having said that the manuscript provides interesting details and prospects for the use of the specific Pseudomonas aeruginosa strain against Botrytis cinerea in tomato plants and a lot of potential against other fungi.
Overall comments
Minor
First of all, the English language should be improved throughout the manuscript. I can provide certain corrections now and after potential revisions to improve the quality of the writing. I only include some of them. Rephrase of many sentences should be conducted.
Also, italics in species, genera, and genes should be checked throughout the manuscript including legends of Figures, Tables, and supplemental materials. There are many inconsistencies that should be improved.
Brief statistics methodology in every table legend should be introduced. Also, statistics should mention the comparison between treatments and/or control groups to help the reader understand the significance of the result findings. The same for supplemental materials.
Also, check superscripts!! (i.e. lines 342, 121 T should be italized)
Major
The manuscript seems unreferenced. There are only 6 references in discussion and only 1 of them regarding previous in vitro or in vivo Pseudomonas aeruginosa strains as biocontrol agents in agriculture. This is my main concern for the manuscript and should be majorly improved, as in my knowledge there are several references that could be incorporated in the discussion and introduction regarding Pseudomonas aeruginosa species.
16s rRNA sequence should be deposited in Genbank or at least introduced as supplemental material for everyone to be accessible. Also in my opinion showing the identification of the bacterial strain with a figure incorporating a molecular technique should be part of the main text and not supplemental material. It is far more interesting and reliable than the colony formation and smell (see later comment as well).
Specific comments
Abstract and Introduction
The introduction is robust and contains a lot of information. Contains 23 references which are adequate. My only concern is that it does not contain any references regarding Pseudomonas aeruginosa as a biocontrol strain against phytopathogenic fungus in the past.
Line 20 antagonistic not antagonism
Line 21 on other
Line 26 in vitro italics (check the whole manuscript)
Line 29 enzymatic
Line 33 perhaps authors should avoid absoluteness. As I discuss later do not forget that P. aeruginosa is also an animal and human pathogen. Authors should only propose their novel isolate and its potentials.
Line 77 a gram
Line 79 perhaps should be written as The Palleroni method or Palleroni et al.,
Results
Results could be more descriptive for some figures for the readers to be able to extract information without having to read a table.
Authors claim that the most identical strain with their strain was Pseudomonas aeruginosa JCM 5962T, (BAM01000316). At first, I was thinking that the accession number inside the parenthesis corresponds to NCBI accession number. I tried to retrieve it and could not. This strain corresponds to a Japanese bacterial library (which is not mentioned anywhere in the manuscript) and the accession number inside the parenthesis does not correspond to anything that I could find. Perhaps the author should rephrase and include the GenBank accession number of the genome or the 16s rRNA accession number of the specific nucleotide sequence.
Table 1. Check italics. The statistics are confusing. Is it contrary to control or something else?
Line 111 smell of colonies is not a scientific method identifying bacterial species. Perhaps should be deleted.
Line 121 T should be superscripted
Lines 120-123 16s accession number should be included
Figure 3 italics on in species. What about statistics? Are they compared to control?
Discussion
Discussion contains a lot of uncited information. In total contains only 6 references which is not a desirable number. Also lines 258-263 seem irrelevant to the scope of the study.
Additionally, another thing is that generally Pseudomonas aeruginosa is a human and animal pathogenic bacterium with its antibiotic resistance being a major threat worldwide. I was expecting authors at least to discuss (if not present results from their strain) this issue as colleagues have done in other works in the past. Virulence study against mammals of the presented strain should be in the future plans of the group.
Line 245 Which studies??
Materials and methods
Statistic methods are not presented.
Lines 302-311 Where did the authors found these primes. Did they design them by themselves or is there any reference that should be included?
There is no THE CTAB METHOD. There is a method incorporating a CTAB-based buffer. Still, the methods of this procedure should be detailed written.
What sequence methodology was incorporated. At least briefly (sanger? Illumina?)
Phylogenetic analysis with MEGA 6.06 needs detailed methods of how was the phylogenetic tree constructed. Also, the company and reference of MEGA 6.06 manufacturer should be mentioned.
Round 2
Reviewer 1 Report
Overall you upgraded the manuscript following reviewers' comments.
Finally, I strongly suggest to check style again.
Your manuscript too much miss-styling, miss-spellings and the lacks of space to allow.
Your research and is basically good, but the low description level makes bad impression and it also makes us doubt your research accuracy.
I would like you to note the importance of description in your mind.
Author Response
请参阅附件。

Reviewer 2 Report
Thanks for the answers and the improvement of the manuscript.
In section 2.6, the name B. cinerae is misspelled (should be B. cinerea).
Point 4.9 - there is still no detailed description of the method used, sample preparation.
References still need to be refined (letter size, no abbreviations, year of publication in different places)
Reviewer 3 Report
The authors of the presented manuscript have done a lot of effort to majorly revise their manuscript, submitting an overall more quality manuscript, reflecting their main results more clearly.
Still, significant amendments need to be met in order for the manuscript to be considered for publication.
My main concern is that the manuscript still lacks the accession number in NCBI of the 16s sequencing of the novel Pseudomonas isolate. (Line 378). This needs to be fixed prior to publication.
Discussion
Lines 298-299 the phrase needs to be revised. Repeatance of green house pot.
Lines 302-303. As said before, Pseudomonas aeruginosa is an animal and human severe pathogen. Doing trials in fields, even if that is a future study, without prior checking its virulence against mamals, is a far-fetched prospect that only creates confusion.
Line 322 requires rephrasing. I do not have a problem starting a sentence with "but", but in my opinion authors could icorporate that statement in previous phrase.
